# A Consistent Method for Generating Synthetic Tabular Data with a Fully Connected Neural Network

**Andrew I. Perminov**
Ivannikov Institute for System Programming of the Russian Academy of Sciences
Moscow, Alexander Solzhenitsyn st., 25.
perminov@ispras.ru

**Denis Y. Turdakov**
Ivannikov Institute for System Programming of the Russian Academy of Sciences
Moscow, Alexander Solzhenitsyn st., 25.
turdakov@ispras.ru

**Andrey P. Kovalenko**
Ivannikov Institute for System Programming of the Russian Academy of Sciences
Moscow, Alexander Solzhenitsyn st., 25.
a.p.kovalenko@ispras.ru

## Abstract

To generate synthetic tabular data for subsequent use in machine learning, it is usually proposed to use all sorts of autoencoders, based on the assumption that their ability to "reproduce" input data from points of a low-dimensional latent space automatically means "reproducing" the statistical and structural properties of the distribution of the original sample. No evidence is provided for the truth of this assumption. The article proposes a consistent data generation method based on the authors' approach to solving the unary classification problem by a fully connected neural network (multilayer perceptron) with piecewise-linear activation functions. The output of such a network is shown to be an adaptive histogram estimate of the distribution density specified on a compact set. Consistency conditions for nonparametric estimates of this type were obtained in Devroye et al. (2013). The tabular data are synthesized by thinning random vectors uniformly distributed on a compact set according to the empirical distribution density obtained. The results of the method are illustrated by model examples.

## 1 Introduction

The generation of synthetic tabular data is an essential component in the development of artificial intelligence (AI) systems, particularly in cases where access to real-world data is restricted due to privacy concerns, proprietary limitations, or data scarcity. Synthetic data enables model training, data augmentation, reproducibility of research, and secure data sharing. To be useful, such data must preserve the structural and statistical properties of real datasets while ensuring the protection of sensitive information.

Synthetic data can be generated using various approaches, including machine learning (ML)-based methods such as generative adversarial networks (GANs) and variational autoencoders (VAEs), as well as traditional statistical methods such as nonparametric density estimation Akkem et al. (2024). Statistical methods typically require consistency as a necessary condition, ensuring that the synthetic distribution converges to the true data distribution as the sample size increases. In contrast, neural network-based generative models often rely on empirical validation using benchmark datasets without theoretical guarantees regarding their ability to preserve statistical properties.

Among the widely used ML-based generative approaches, autoencoders attempt to reconstruct the original data from a compressed latent space, assuming that such a transformation captures the essential features of the distribution. GANs employ a generator-discriminator framework to produce samples that resemble real data Jordon et al. (2018), while VAEs introduce probabilistic modeling to learn a latent representation that enables sample diversityWan et al. (2017). Despite their success, these methods suffer from several drawbacks, such as mode collapse in GANs, difficulties in defining an appropriate latent space in autoencoders, and the challenge of maintaining statistical consistency in VAEs. Moreover, none of these methods offer a theoretical foundation ensuring the preservation of the statistical properties of the original data.

In this paper, we propose a consistent method for synthetic data generation using unary classification with a fully connected neural network (multilayer perceptron, MLP) equipped with piecewise-linear activation functions (ReLU, Leaky-ReLU, Abs). Unlike traditional generative models, our approach employs a trained classifier to approximate the density of the original data distribution. Specifically, we train an MLP to distinguish real data points from a uniform background distribution within a compact domain. The classifier's output is then used to filter newly sampled background points, effectively producing a synthetic dataset that follows the empirical density of the original data.

Neural networks have previously been applied to density estimation, as discussed in (Magdon-Ismail & Atiya, 1998). However, unlike traditional neural density estimators, which approximate probability densities directly via parameterized functions, our approach utilizes an MLP in a classification framework, interpreting its output as an adaptive histogram estimator. This interpretation allows for direct connections to nonparametric density estimation methods and provides a structured approach to synthetic data generation through controlled sampling.

Our key hypothesis is that the trained neural network acts as an adaptive histogram estimator of the underlying density function. Since MLPs with piecewise-linear activation functions partition the feature space into linear regions, they naturally approximate complex distributions. This formulation aligns with nonparametric density estimation techniques, particularly histogram-based methods, whose consistency conditions have been established in Devroye et al. (2013). While traditional histogram estimators introduce discontinuities, the neural network provides a smooth approximation, as it adjusts hyperplane orientations to balance density variations.

The main contributions of this work are as follows:

- We introduce a consistent synthetic data generation method based on unary classification with a multilayer perceptron.

- We provide a theoretical perspective on how MLPs approximate density functions by partitioning the input space into linear subregions.

- We demonstrate empirically that the proposed method preserves the cluster structures and statistical properties of the original data distribution.

- We present visualizations of synthetic datasets and covariance matrix comparisons for high-dimensional cases, confirming the validity of the approach.

The remainder of the paper is organized as follows. Section 2 presents the formal problem statement and methodology. Section 3 describes the experimental setup and provides results on synthetic datasets. Section 4 discusses the theoretical implications of the method and its limitations. Finally, Section 5 concludes the paper and outlines directions for future research.

## 2 PROBLEM STATEMENT AND METHODOLOGY

### 2.1 PROBLEM STATEMENT

Let $X = \{x_1, x_2, \ldots, x_n\} \subset \mathbb{R}^d$ be a given dataset sampled from an unknown probability distribution with a density function $p_X(x)$. The goal of synthetic data generation is to construct a new dataset $\tilde{X} = \{\tilde{x}_1, \tilde{x}_2, \ldots, \tilde{x}_m\}$ that approximates the statistical properties of $X$ while maintaining privacy constraints.

A common approach to density estimation involves constructing a nonparametric estimator $\hat{p}_X(x)$ of $p_X(x)$. We propose an alternative method based on unary classification in which a neural network is trained to distinguish real data from a background distribution.

## 2.2 UNARY CLASSIFICATION

The method for generating synthetic samples is based on the construction of a Bayesian unary classifier. In (Lukianov et al., 2024), a method for extrapolating a Bayesian binary classifier was proposed, where an additional artificially generated "background" class with a label "0" is introduced along-side two classes labeled "+1" and "-1". This background class represents a random sample drawn from a uniform distribution over a given compact set. In this formulation, the modified classifier can not only assign an observation to one of the two classes but also reject classification if the discriminant function is close to zero. As a result, input observations falling outside the support of the original distribution will be rejected.

A unary Bayesian classifier differs from the modified binary classifier in that the original dataset consists of observations from only one class, labeled "1," while the background class observations are labeled "0." Formally, the unary classification problem is formulated as follows.

Let $(X, Y)$ be a random variable where $X$ is a $d$ - dimensional random vector with a mixture density $\alpha f(x) + (1-\alpha)p(x)$, where $f(x)$ is the density of the target class, $p(x)$ is the density of the uniform background distribution over a compact set $K$, and $\alpha$ is a weighting coefficient, $0 \leq \alpha \leq 1$. The label $Y$ takes values of 1 or 0 depending on whether $X$ belongs to the original sample or the background.

The posterior probability, or regression function, of $Y$ given $X$ is defined as:

$$g(x) = P(Y = 1|X = x) = E(Y|X = x) = \frac{\alpha f(x)}{\alpha f(x) + (1-\alpha)p(x)}. \tag{1}$$

If $g(x)$ were known, the unary classification problem could be solved by classifying $x$ as part of the target distribution if $g(x) > 0$, and rejecting it otherwise. However, since $g(x)$ is typically unknown, an approximation must be constructed from the given data.

Let $c(X)$ be a continuous function defined on $K$. Consider the mean squared approximation problem:

$$c^*(x) = \arg\min E(c(x) - Y)^2. \tag{2}$$

Since:

$$E(c(x) - Y)^2 = E(c(x) - g(x) + g(x) - Y)^2 = E(c(x) - g(x))^2 + E(g(x) - Y)^2, \tag{3}$$

and the second term is independent of $c(x)$, the problem reduces to the approximation of the regression function:

$$c^*(x) = \arg\min E(c(x) - g(x))^2. \tag{4}$$

As the function $c(x)$, a multilayer perceptron (MLP) with $L$ hidden layers of $k$ neurons each and piecewise linear activation $abs$ is considered. According to the universal approximation theorem (Cybenko, 1989), for any $\varepsilon > 0$, there exist values of $k$ and $L$ such that for any $x \in K$:

$$\sup |c(x) - g(x)| < \varepsilon. \tag{5}$$

Thus, an $\varepsilon$ - approximate solution of (2) theoretically exists.

For a statistical formulation, let the given sample $\{X_i, Y_i\}_{i=1}^n$ where $X_i \in K$ be interpreted as a labeled set of $n$ observations from the target density $f(x)$. To construct a mixed dataset with density

$\alpha f(x) + (1 - \alpha)p(x)$, artificial background samples $\{X_j, 0\}_{j=n+1}^{n+m}$ are added, where $m = n \cdot \frac{1-\alpha}{\alpha}$ and $X_j$ is drawn from a uniform distribution over $K$.

Let $C(k, L)$ be the class of MLPs with piecewise linear activation $abs$ and given $L$ and $k$. The optimization problem is formulated as:

$$\sum_{i=1}^{n+m} (c_n(X_i) - Y_i)^2 \to \min, \tag{6}$$

where the minimization is over all $c_n(X) \in C(k, L)$.

Let $c^*(X)$ be the solution (6), referred to as the neural regression function. The corresponding perceptron partitions $K$ into $N$ disjoint cells: $K = \{K_1, K_2, \ldots, K_N\}$ (Kovalenko, 2022).

To justify consistency of $c^*(x)$, consider a piecewise constant function of histogram regression $h_n(X)$ defined by minimizing:

$$\sum_{i=1}^{n+m} (h_n(X_i) - Y_i)^2 \to \min, \tag{7}$$

over all piecewise constant functions defined on cells $K_r$ of $K$. Let $X \in K_r$. Within each cell $K_r$, the problem (7) reduces to:

$$n_1(X) \cdot (h_{nr} - 1)^2 + n_0(X) \cdot (h_{nr} - 0)^2 \to \min, \tag{8}$$

where $h_n(X) = h_{n_r}, n_1(X) = \sum_{i=1}^{n+m} I_{\{X_i \in K_r, Y_i=1\}}, n_0(X) = \sum_{i=1}^{n+m} I_{\{X_i \in K_r, Y_i=0\}}$. Differentiating with respect to $h_{nr}$ yields the solution of (7):

$$h_n^*(X) = \frac{n_1(X)}{n_1(X) + n_0(X)} = \frac{f_n(X)}{f_n(X) + \frac{1-\alpha}{\alpha} \cdot p_n(X)}, \tag{9}$$

where $f_n(X) = \frac{n_1(X)}{n \cdot V(K_r)}$ - is histogram density $f(x)$ estimates at cell $K_r$, $p_n(X) = \frac{n_0(X)}{n \cdot V(K_r)}$ - is histogram uniform density estimates at cell $K_r$ and $V(K_r)$ - is measure of the cell $K_r$.

Asymptotic conditions for strong consistency of adaptive histogram estimators are given in (Devroye, 1989; Devroye et al., 2013), requiring cell diameters to shrink while maintaining sufficiently many points per cell. These conditions hold even for moderate values of $k$ and $L$, e.g., for $d = 10$, $k = 10$, $L = 2$, $N$ exceeds tens of thousands, necessitating millions of background points for proper coverage. In filled cells, where both target and background points are present, $h_n^*(X)$ and $c^*(X)$ are close. In background-only cells, $h_n^*(X) = 0$, while $c^*(X)$ is interpolated due to continuity. In high-density regions, $c^*(X)$ is significantly above zero, whereas in low-density regions, it approaches zero.

Therefore, for sufficiently large $n$ and appropriately chosen $k$ and $L$, the neural regression $c^*(X)$ is a consistent estimator of $g(X)$.

### 2.2.1 PARTITIONING INDUCED BY THE PERCEPTRON

A multilayer perceptron with piecewise linear activations partitions the input space into disjoint regions, analogous to histogram bins. Each region is defined by a unique pattern of neuron activations: if the sign of each neuron's pre-activation output is fixed, the perceptron behaves as a linear operator within that region. Given a cell, the density estimate follows:

$$p(x) \approx \frac{N_{\text{data}}}{N_{\text{data}} + N_{\text{background}}}, \tag{10}$$

where $N_{data}$ and $N_{background}$ denote real and background sample counts.

Unlike traditional histograms with sharp bin boundaries, the perceptron forms a continuous approximation, as its activation patterns define smooth transitions between regions. Figure 1 illustrates this structure using a simple 2-5-5-1 network applied to a spiral dataset. The decision tree representation clarifies how hierarchical neuron activations contribute to partition formation (showed only neurons that split space, neurons A2, A4, A3, and C1 exhibit negative values, while neurons A1 and A3 are positive).

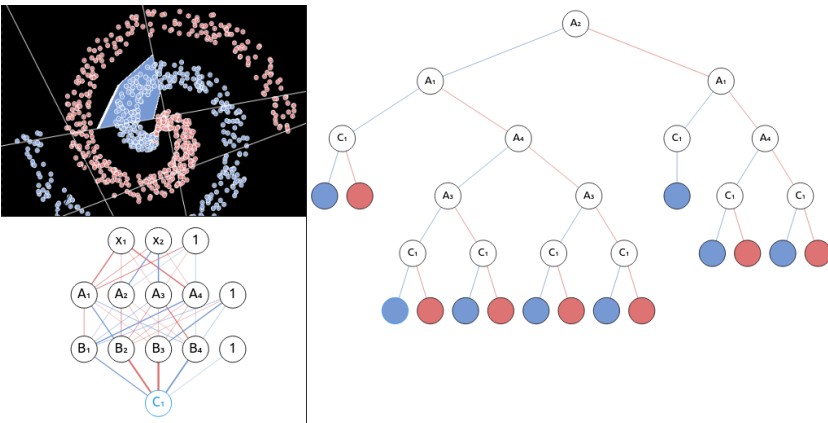

Figure 1: Example of perceptron decision tree based on neurons signs.

This theoretical foundation suggests that the classifier-based synthetic data generation method effectively combines the strengths of both histogram-based and continuous density estimation approaches, allowing for structured yet smooth synthetic data generation.

## 2.3 METHODOLOGY

### 2.3.1 BACKGROUND DATA GENERATION

The background points are sampled from a uniform distribution within a compact domain $K \subset \mathbb{R}^d$. For each dataset, $K$ is chosen as an axis-aligned hyperrectangle that extends beyond the real data distribution by a margin of 20–50% in each dimension. This margin ensures that the classifier receives sufficient negative samples to distinguish the support of the data distribution and allow for the decision function to be pushed toward the zero plane. A set of background points $B = \{b_1, b_2, \ldots, b_n\}$ is sampled uniformly from $K$, ensuring that $|B| = |X|$.

### 2.3.2 TRAINING THE CLASSIFIER

A multilayer perceptron (MLP) classifier $c : \mathbb{R}^d \to [0, 1]$ is trained on the combined dataset $X \cup B$ with binary labels:

$$c(x) = 1, \quad x \in X,$$
$$c(b) = 0, \quad b \in B.$$

The network is optimized to minimize the mean squared error (MSE) loss:

$$L = \sum_{x \in X} (1 - c(x))^2 + \sum_{b \in B} (0 - c(b))^2. \tag{11}$$

Background points are generated at each epoch during training, rather than being fixed once before the start of the training process.

The model is trained using the MSE loss rather than cross-entropy. While cross-entropy is a standard choice for classification, MSE provides a smoother approximation of posterior probabilities without requiring an explicit softmax or sigmoid transformation and does not enforce a hard decision boundary. This property aligns with our goal of estimating a continuous density function, where

the network output should reflect a smooth probability estimate rather than a sharp classification decision. Additionally, MSE implicitly encourages regression-like behavior, allowing the output to approximate $g(x)$ without requiring explicit probabilistic normalization and aligns better with the histogram interpretation of the classifier output.

### 2.3.3 SYNTHETIC DATA SAMPLING

Once trained, the classifier is used to filter new background samples. A set of candidate points $\tilde{B}$ is drawn uniformly from $K$, and each point $\tilde{b} \in \tilde{B}$ is retained with probability $c(\tilde{b})$. The resulting set $\tilde{X}$ serves as the synthetic dataset:

$$\tilde{X} = \{\tilde{b} \in \tilde{B} \mid \xi < c(\tilde{b})\}, \tag{12}$$

where $\xi \sim \mathrm{Uniform}(0, 1)$ is a random variable.

## 3 EXPERIMENTAL SETUP AND RESULTS

### 3.1 EXPERIMENTAL SETUP

To evaluate the effectiveness of the proposed synthetic data generation method, experiments are conducted on synthetic datasets where the underlying data distribution is explicitly known. This allows for an objective assessment of the method's ability to preserve statistical properties and structural characteristics.

Three distinct types of datasets are considered:

- **Spiral dataset**: a two-dimensional dataset where points form a spiral pattern (Figure 2 on the left). This dataset tests the model's ability to capture and replicate complex cluster structures with nonlinear boundaries.

- **Two-sphere dataset**: a two-dimensional dataset consisting of two circular clusters with radius $R = 0.25$ positioned at $(-0.5, 0)$ and $(0.5, 0)$, separated by a distance $d = 0.5$ along the $X$-axis (Figure 2 on the right). This dataset is designed to test the method's ability to maintain well-separated clusters in the generated data.

- **Gaussian mixtures**: multivariate Gaussian distributions with known means and covariance matrices (Figure 3). This dataset is used to assess whether the synthetic data preserves high-order statistical dependencies, such as covariance structure.

For each dataset, the classifier-based generation method is applied, and the resulting synthetic dataset is compared to the original data using both visual and statistical analyses.

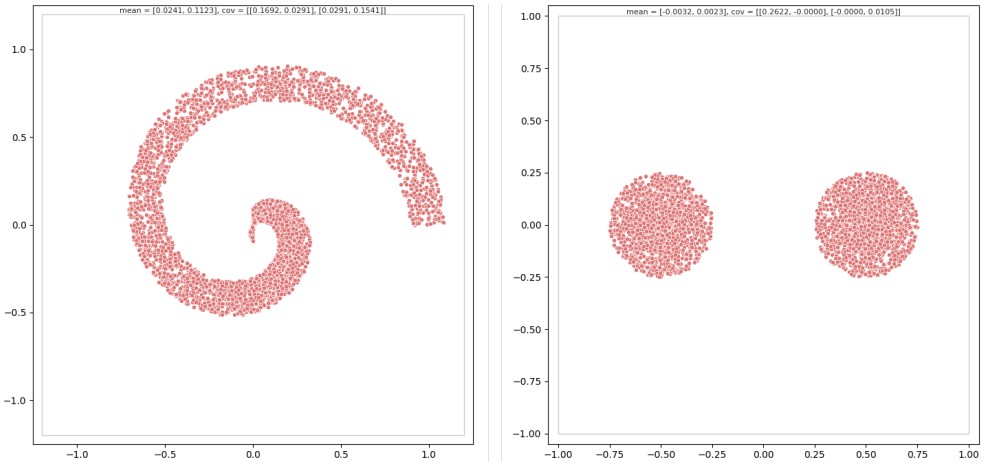

Figure 2: Spiral dataset (left) and spheres dataset (right) examples.

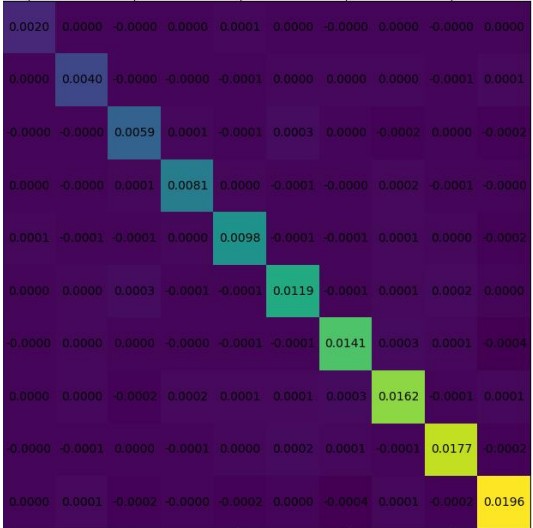

Figure 3: Gaussian dataset covariance matrix example.

To assess the robustness of the approach, multiple neural network architectures are employed for training:

- **d-10-1**: A simple architecture with a single hidden layer of 10 neurons.
- **d-10-100-1**: A deeper network with an intermediate layer of 100 neurons to increase capacity.
- **d-10-10-10-1**: A balanced architecture with three hidden layers of 10 neurons each.

where $d$ represents the input dimensionality of the dataset.

Training is conducted for 100 epochs using a batch size of 32. The datasets consist of 1000 points. Mean Squared Error (MSE) loss is employed instead of cross-entropy to provide a smooth output distribution that aligns with density estimation objectives. The models are optimized using the Adam optimizer with a learning rate of $10^{-3}$.

### 3.2 RESULTS

The experimental results demonstrate that the proposed approach effectively replicates the structure of the original datasets while maintaining key statistical properties.

For the spiral dataset, visual inspection of the generated synthetic data (Figure 4) shows that the method successfully captures the intricate, nonlinear cluster structure. The synthetic points align well with the original spiral arms, indicating the classifier effectively models the density of the dataset.

For the two-sphere dataset, the generated synthetic data maintains the separation between the two clusters (Figure 5). The density distribution of the synthetic points remains consistent with that of the original dataset, demonstrating that the proposed method preserves cluster integrity in settings where distinct modes are present.

For the Gaussian mixtures, the covariance matrices of the synthetic and original datasets are computed and compared (Figure 6). The results indicate that the synthetic dataset closely matches the covariance structure of the original data. However, for high-dimensional Gaussian distributions (e.g., $d = 10$), while the overall structure of the covariance matrix is preserved, the variance values tend to be slightly inflated. This effect is attributed to noise amplification in high-dimensional spaces, where the classifier's decision boundaries become more fragmented due to sparsity.

To further illustrate the quality of the generated data, Figure 7 presents scatter plots of synthetic and original samples projected onto each pair of feature dimensions. In these plots, red points represent

real data, while green points denote synthetic samples. The visual comparison confirms that the synthetic data retains the cluster structure of the original distribution, although slight deviations in density can be observed in some projections.

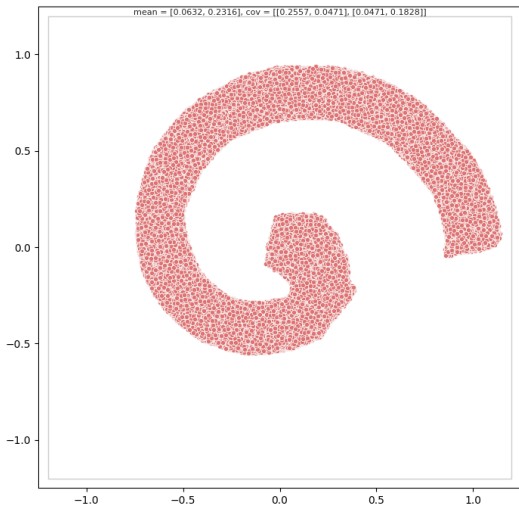

Figure 4: Spiral synthetic data.

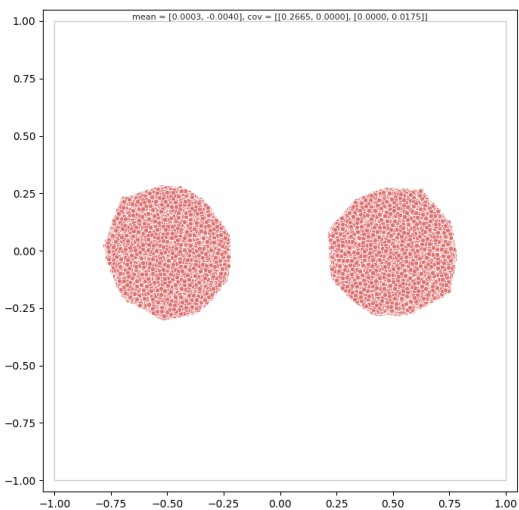

Figure 5: Spheres synthetic data.

To provide a comprehensive overview of the method's performance, we present results for the most representative neural network configurations. While smaller models (e.g., **d-10-1**) demonstrate a basic ability to separate density regions, deeper architectures (e.g., **d-10-100-1**) offer enhanced fidelity in capturing fine-grained structures within the data. The trade-offs between model complexity and generalization to different distributions are further discussed in Section 4.

Overall, the results confirm that the proposed classifier-based synthetic data generation method can effectively model diverse data distributions, maintaining both global and local statistical properties.

## 4 DISCUSSION

Despite the effectiveness of the proposed approach, challenges persist, particularly concerning high-dimensional data. As the dimensionality increases beyond 10, generating high-quality synthetic

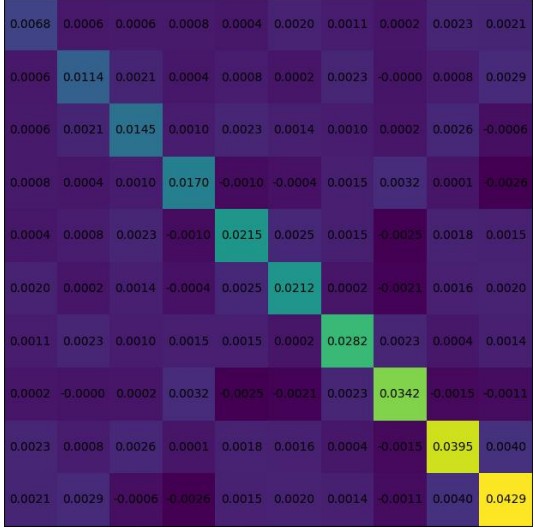

Figure 6: Gaussian synthetic data covariance matrix.

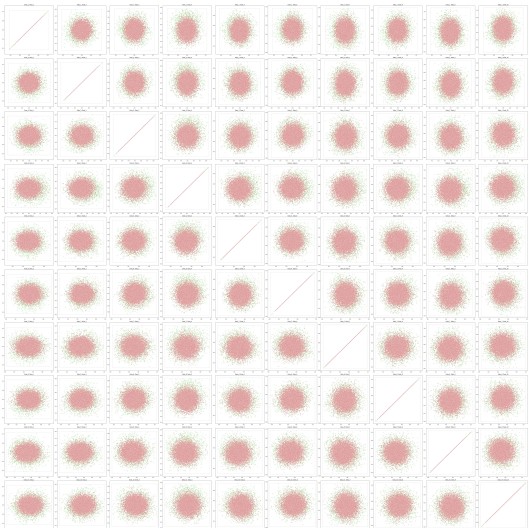

Figure 7: Gaussian synthetic data projections.

background samples becomes increasingly difficult. This is due to the sparsity of data in high-dimensional spaces, which affects the classifier's ability to generalize well across the entire domain.

Another critical observation is the effect of network depth and neuron count on the partitioning of the input space. Larger networks introduce a finer subdivision of the feature space, leading to smaller partitions analogous to histogram bins. This behavior suggests that as the network complexity grows, the generated density function exhibits finer granularity, which may not always align with the desired statistical properties of the target distribution.

Additionally, rather than directly sampling from the generated probability distribution, it may be beneficial to introduce a confidence threshold $\beta$. By filtering out points where $C(x) < \beta$, it is possible to ensure that only highly confident synthetic samples are retained. This threshold acts as a tunable parameter that balances the trade-off between dataset quality and sample size: increasing $\beta$ reduces noise in the generated dataset but at the cost of lower sample diversity.

Future work could explore adaptive thresholding mechanisms and alternative techniques for handling high-dimensional distributions more effectively.

### 4.1 DIFFERENCES FROM LATENT SPACE-BASED MODELS

Unlike deep generative models such as CTGAN (Habibi et al., 2023) and TVAE (Ishfaq et al., 2018), which model latent representations of tabular data, our method is designed as a direct density and adaptive histogram-based approach. The primary focus of this work is on the theoretical formulation of density estimation via unary classification rather than empirical performance on structured tabular datasets. Future work could explore comparisons with these models in real-world applications.

### 4.2 BIAS PROPAGATION AND ETHICAL CONSIDERATIONS

While synthetic data generation can help mitigate data scarcity and privacy concerns, it also carries risks of bias propagation. Since the proposed method learns the density distribution from an existing dataset, any biases present in the original data may be reflected in the generated samples. This effect is particularly relevant when the training data exhibit class imbalances or underrepresented subpopulations. Unlike adversarial generative models, which can explicitly enforce fairness constraints, our approach relies on the assumption that the classifier approximates the true underlying distribution without correction mechanisms.

To address this limitation, future work could explore methods for bias detection and mitigation within the proposed framework. Potential strategies include modifying the background sampling process to compensate for imbalanced regions or incorporating fairness-aware training objectives to adjust the classifier's density estimates.

## 5 CONCLUSION

The proposed classifier-based synthetic data generation method provides a structured approach to generating high-fidelity synthetic datasets while preserving key statistical properties of the original data. Through extensive experiments on various synthetic datasets, including spirals, separated spherical clusters and Gaussian mixtures, the approach has demonstrated its capability to maintain both local and global data structures.

One of the key advantages of this method is its ability to adaptively shape synthetic data distributions based on the learned classifier output. By leveraging neural networks with different architectures, the model is capable of capturing complex density patterns, ensuring that the synthetic data remains a faithful representation of the original distribution. However, as highlighted in the discussion, certain limitations emerge, particularly in high-dimensional settings, where the generation of meaningful background samples remains a challenge.

The experimental findings indicate that the choice of network depth and width significantly impacts the granularity of the generated density function. More complex networks tend to create finer partitions in feature space, which can be beneficial for capturing intricate details but may also lead to over-segmentation of density regions. Furthermore, the introduction of a confidence threshold $\beta$ offers a mechanism to refine the selection of synthetic samples, providing a balance between dataset fidelity and sample sufficiency.

Future work should explore strategies to mitigate the challenges associated with high-dimensional spaces, potentially incorporating adaptive sampling techniques or hybrid approaches that combine classifier-based generation with density estimation methods. Additionally, extending the method to real-world datasets and assessing its effectiveness in privacy-preserving data synthesis remains an important direction for further research.

In conclusion, the proposed approach represents a step forward in synthetic data generation, offering a practical and scalable framework for preserving statistical characteristics while allowing for controlled dataset synthesis. Its flexibility make it a promising candidate for applications in machine learning, privacy preservation, and statistical modeling.

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
