# OpenReview forum: "A Consistent Method for Generating Synthetic Tabular Data with a Fully Connected Neural Network"
_mathai.club/MathAI/2025/Conference — MathAI 2025 Oral_

### Official Review · Reviewer_e6Lj · 2025-02-27
**A Consistent Method for Generating Synthetic Tabular Data with a Fully Connected Neural Network**

**Rating:** 8
**Confidence:** 4

**Review:**

The authors propose a method for generating high-quality synthetic datasets that preserve the main statistical characteristics of the original data. This classifier-based method allows adaptively generating synthetic data distributions based on the output of a trained classifier. Using neural networks with different architectures, the model is able to handle complex density patterns. Experiments conducted on a number of synthetic datasets, such as spirals, individual spherical clusters, and Gaussian mixtures, showed that this method effectively preserves both local and global data structures.

---

### Official Review · Reviewer_Rkt4 · 2025-02-27
**This paper proposes a novel method for synthetic tabular data generation using MLPs with piecewise-linear activations. Experiments on synthetic datasets demonstrate the method's ability to preserve statistical properties, though the evaluation lacks real-world validation and comparative benchmarking against existing generative models**

**Rating:** 6
**Confidence:** 3

**Review:**

### 1. **Summary**

The paper proposes a method for generating synthetic tabular data using a fully connected neural network (MLP) with piecewise-linear activation functions. The authors frame the problem as a unary classification task, where the MLP distinguishes real data from a uniform background distribution. The classifier’s output is interpreted as an adaptive histogram estimator of the data density, leveraging consistency results from nonparametric density estimation. Synthetic data is generated by thinning uniformly sampled points using the classifier’s probability estimates. Experiments on synthetic datasets (spiral, two-sphere, Gaussian mixtures) demonstrate the method’s ability to preserve statistical properties, such as covariance structures and cluster separation.

---

### 2. **Strengths and Weaknesses**

#### **a. Originality**

- **Strengths**: The work introduces a novel application of MLPs for density estimation, diverging from traditional generative models (e.g., GANs, VAEs). The connection between MLP-based classification and adaptive histogram estimation is creative, with theoretical grounding in nonparametric statistics.

- **Weaknesses**: While the combination of MLPs and histogram methods is new, the paper could better differentiate itself from prior neural density estimation techniques (for example, see Magdon-Ismail, Malik and Amir F. Atiya. "Neural Networks for Density Estimation" in Neural Information Processing Systems, 1998) and clarify how the proposed method improves upon them.

#### **b. Quality**

- **Strengths**: Experiments on synthetic datasets validate the method’s ability to replicate cluster structures and covariance matrices.

- **Weaknesses**: The evaluation is limited to synthetic data; real-world benchmarks are absent. The claim of "consistency" is not empirically validated for large N, and the impact of network architecture (depth/width) on performance is underexplored.

#### **c. Clarity**

- **Strengths**: The problem statement and methodology are logically structured.

- **Weaknesses**: The training process (e.g., generating of background points) needs clearer exposition.

#### **d. Significance**

- **Strengths**: The method provides a theoretically grounded alternative to black-box generative models, with potential applications in privacy-preserving data synthesis. The adaptive partitioning via MLPs is a promising direction.

- **Weaknesses**: Practical significance is limited without validation on real-world datasets. The scalability to high dimensions (d > 10) is discussed but not rigorously tested.

---

### 3. **Questions for the Authors**

1. **Theoretical**: How does the choice of MSE loss (over cross-entropy) impact the density estimation quality? Is there a theoretical justification for this choice?

2. **Experiments**: Why were no comparisons made with existing synthetic data generators (e.g., CTGAN, TVAE)? Could the method be tested on real-world tabular datasets (e.g., UCI repositories)?

3. **Implementation**: How is the compact domain K precisely defined (e.g., margins for different datasets)? Does the 20–50% margin affect results significantly?

---

### 4. **Limitations**

- The paper acknowledges high-dimensional challenges but does not quantify performance degradation (e.g., via dimensionality scaling experiments).
- Limitations in handling sparse regions and computational costs for large d are noted but not mitigated.
- **Suggestion**: Include a sensitivity analysis for the margin parameter and evaluate runtime vs. dimensionality.

---

### 5. **Ethical Concerns**

No major ethical issues are apparent. However, the authors should discuss potential misuse of synthetic data (e.g., bias propagation) and privacy risks if the method is applied to sensitive datasets.

---

### 6. **Soundness**

The technical claims are supported by theory and synthetic experiments, but the lack of real-world validation and scalability analysis limits confidence.

---

### 7. **Presentation**

The core ideas are clear, but incomplete methodological details reduce clarity.

---

### 8. **Contribution**

The method is novel and theoretically interesting, but its practical impact remains unproven without broader empirical validation.

---

### 9. **Overall Score**
The paper presents a technically sound and original approach with theoretical merit. However, the limited evaluation scope and presentation flaws weaken its readiness for publication. A more comprehensive evaluation (e.g., real-world data, comparisons) would strengthen its contribution.

---

**Feedback for Rebuttal**: The authors should address the questions above, particularly regarding scalability, loss function choice, and real-data experiments. Clarifying the methodology would enhance readability.

---

---

### Decision · Program_Chairs · 2025-03-08

**Decision:**

Accept (Oral)

**Comment:**

Your article has been accepted and you can give a talk on the article. All articles will be sorted by rating and within the available conference places one author from each article will be invited. If there are not enough places, then you will either have the opportunity to speak remotely or come at your own expense!